# TABGFN:
# TABULAR DATA GENERATION BASED ON GFLOWNETS

## ABSTRACT

Generation of synthetic tabular data plays an important role in privacy-preserving data sharing, training data augmentation, data imputation, and algorithm development in various domains such as healthcare and finance. Achieving both high predictive performance and model traceability in tabular data generation is challenging for neural-network-based algorithms due to their inherent model opacity. To overcome this limitation, we present a novel approach for generating synthetic tabular data called TabGFN. It employs generative flow networks for feature generation and uses the critic network of the Wasserstein generative adversarial network with gradient penalty as its reward function. Through simultaneous and iterative training of the flow network and reward function, TabGFN explores a directed acyclic graph of the generative state space, yielding a generative model that represents conditional relationships and feature order. Benchmark tests on diverse datasets demonstrate that the quality of the synthetic data by TabGFN is superior or comparable to that of state-of-the-art algorithms. Moreover, the entire generation process is traceable, as its individual steps are explicitly provided. This traceability enables the discovery of mutual dependencies between features, leading to an interpretable model, which is crucial for high-stakes decision-making. Thus, the proposed approach offers an effective solution for generating tabular data, providing both high-quality synthesis and traceability.

## 1 INTRODUCTION

Generating synthetic data from tabular data is an active research area. Typically, high-value industries such as healthcare and finance are the main producers and consumers of tabular data. A large portion of tabular data is human-generated and refined, resulting in significant costs, which presents challenges in data creation, collection, and sharing. This has led to a high demand for data augmentation and generation techniques. State-of-the-art (SOTA) neural-network-based techniques, such as the generative adversarial network (GAN) (Goodfellow et al., 2014) and denoising diffusion probabilistic model (DDPM) (Ho et al., 2020), have been applied to tabular data generation, demonstrating success with unstructured data such as images. Notable examples include CTGAN (Xu et al., 2019), medGAN (Choi et al., 2017), and tableGAN (Park et al., 2018), all grounded in the GAN. More recently, a diffusion-based method, TabDDPM (Kotelnikov et al., 2023), has exhibited SOTA performance. However, these approaches have the limitation of the generator functioning as a black-box model, lacking traceability. GANBLR (Zhang et al., 2021), based on both GAN and Bayesian networks, provides interpretability but its generative performance does not match that of SOTA models.

The features constituting tabular data inherently contain more condensed information than those constituting unstructured data. Therefore, there have been efforts to interpret models to understand the interactions between features. A notable example is the model-agnostic interpretation method SHAP (Lundberg & Lee, 2017). Additionally, SOTA predictive models such as XGBoost (Chen & Guestrin, 2016) and LightGBM (Ke et al., 2017) provide feature importance by analyzing the decision tree splits. Although the classic graphical model, such as the Bayesian network, may underperform compared with various modern machine learning algorithms, its inherent interpretability has led to ongoing proposals for methods to learn the optimal structure (Kitson et al., 2023).

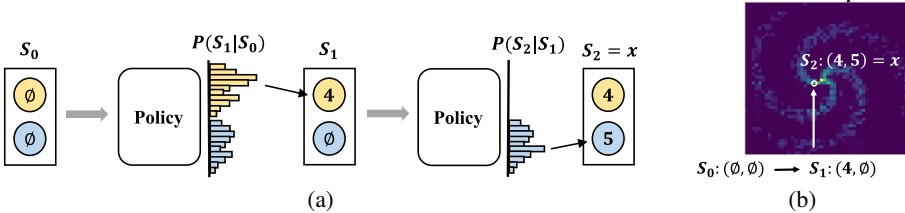

Figure 1: Overview of generation process in TabGFN. The generation procedure of spiral data points distributed in a two-dimensional plane is used as an example. (a) Explicit feature generation process of TabGFN, where the policy provides the probabilities of the next feature. (b) Feature determination process in the reward space, where a high-reward mode is approached through feature determination.

Generative flow networks (GFlowNets) (Bengio et al., 2021) have recently emerged as a groundbreaking generative model, demonstrating achievements in domains such as molecule generation (Bengio et al., 2021), DNA sequence generation (Jain et al., 2022), and discrete image generation (Zhang et al., 2022). GFlowNets learn a stochastic policy to produce compositional objects such as molecules. The generated objects are sampled with a probability proportional to a given non-negative reward. GFlowNets treat the creation process as a directed acyclic graph (DAG) and interpret the stochastic process along the DAG as a flow of rewards.

GFlowNets can be used to represent a joint distribution as a chain of conditional distributions (Bengio et al., 2023). The state (DAG node) in GFlowNets determines transitions (DAG edges) based on a learned policy as a conditional probability. By considering action sampling for state transitions as sampling a feature and corresponding category, it is possible to modify GFlowNets to capture the conditional distribution between features in tabular data. Furthermore, the transition probabilities in GFlowNets are explicitly provided, enabling traceability in the tabular data generation process (see Figure 1a).

To generate samples over the joint distribution of features, GFlowNets requires information about the joint distribution in the form of rewards. In GFlowNets, the reward acts as the unnormalized probability of creating an object. In the Wasserstein GAN with gradient penalty (WGAN-GP) (Gulrajani et al., 2017), a critic network plays the role of discriminator, serving as a value function to criticize the quality of the generated samples. TabGFN employs it as a reward function and learns to generate samples with probabilities proportional to the scores from the critic network, resulting in the generation of high-quality data.

In this study, we propose TabGFN, a tabular data generation algorithm that employs GFlowNets and the critic network of WGAN-GP to approximate the unnormalized distribution of real data. TabGFN models the conditional distribution between features using GFlowNets and chains them explicitly over the reward space, providing a traceable generation process.

The results of experiments conducted on 26 simulated and real-world datasets indicate that data generated by TabGFN are competitive with data generated by existing generative models. In addition, we provide examples of an interpretation of the generation process to demonstrate the advantages of the traceable generation process of TabGFN.

The main contributions of this study are summarized as follows:

(1) We propose TabGFN, a novel tabular data generation method employing GFlowNets and the critic network of WGAN-GP, and a joint training framework

(2) Competitive benchmark results are obtained by TabGFN across various simulated and real-world datasets.

(3) Demonstrating the traceability of the generation process in TabGFN.

## 2 RELATED WORK

Statistical methods have been proposed based on Bayesian networks (Heckerman, 1997) and SMOTE (Chawla et al., 2002). Recent methods based on neural networks can be divided into two approaches: GAN-based and diffusion-based methods, both of which are prominent in image generation.

### 2.1 GAN-BASED APPROACH

The GAN framework consists of a generator that produces synthetic data and a discriminator that distinguishes generated and real data. The generator aims to produce high-quality data that can deceive the discriminator by adversarial training against it.

GAN has achieved great success in unstructured data generation and has been applied to tabular data. Prominent results include medGAN, which employs an auto-encoder as the generator, and tableGAN, which uses a convolutional neural network in the generator. However, a critical problem with the vanilla GAN framework is its inability to accurately approximate imbalanced feature distributions, with the model often struggling to generate infrequently occurring categories (Xu et al., 2019). CTGAN addresses this problem by using a conditional generator, in which a random categorical feature and its corresponding category are sampled from real data. The generator is constrained to generate new data while preserving the sampled category. For continuous features, CTGAN utilizes a Gaussian mixture model to identify clusters in the given data and then applies a mode-specific normalization method that normalizes differently for each cluster.

In a typical GAN framework, the generator encodes data from random noise, which makes the generator a black-box model. GANBLR considers the importance of interpretability in tabular data generation and employs K-dependence Bayesian estimators (KDB) as its generator model. By exploiting the characteristics of the Bayesian network, it offers an interpretable generative model. However, due to the computational cost of Bayesian network structure learning, there is a limitation on the depth of feature dependencies. Furthermore, the quality of the generated data does not match that of SOTA models.

### 2.2 DIFFUSION-BASED APPROACH

The diffusion-based approach is inspired by the recent success of DDPM in image generation (Ho et al., 2020). DDPM adopts a generative framework motivated by physical processes where particles disperse over time, eventually diffusing into random noise (Sohl-Dickstein et al., 2015). DDPM models this diffusion process by iteratively adding random noise to the original data as time progresses. It also proposes a reverse diffusion process, starting from random noise and aiming to reconstruct the original data. This reverse process unfolds in the form of a Markov chain. Consequently, the reverse process develops the joint distribution of features from random noise. TabDDPM is the most recently proposed method for tabular data modeling using diffusion. It processes continuous and categorical features using the diffusion techniques of vanilla DDPM and multinomial DDPM (Hoogeboom et al., 2021), respectively. Given that DDPM has achieved SOTA performance in image generation, it has also proven to be effective in tabular data generation. However, although TabDDPM can generate high-quality data, it faces challenges related to traceability.

## 3 BACKGROUND

### 3.1 GFLOWNETS

GFlowNets are a generative model over a DAG $\mathcal{G} = (\mathcal{S}, \mathcal{A})$. Nodes $\mathcal{S}$ represent partially constructed objects (states), including a unique initial state $s_0$ and set of terminal states $\chi$. Edges $\mathcal{A}$ correspond to stochastic state transitions (actions) over the state. The leaf nodes of the DAG are the target objects to be generated, denoted by $x = s_n \in \chi$. Each target object $x$ is associated with a reward obtained from a fixed value function $R : \chi \mapsto \mathbb{R} \geq 0$. A trajectory $\tau$ from the initial state $s_0$ to the terminal state $s_n$ is expressed as follows: $\tau = (s_0 \rightarrow s_1 \rightarrow \cdots \rightarrow s_n)$. The transition from the non-terminal state $s_i$ to the next state $s_{i+1}$ is represented as $P_F(s_{i+1}|s_i)$, and the generation probability of $\tau$ is

formulated as follows:

$$P_F(\tau) = P_F(s_1|s_0)P_F(s_2|s_1)\cdots P_F(s_n|s_{n-1}). \tag{1}$$

In this context, the transition probability $P_F$ is referred to as the forward policy.

The objective of GFlowNets is to generate a target object $x$ with a probability proportional to the magnitude of a given reward, that is, $P(x) \propto R(x)$. GFlowNets consider the initial state of the DAG as the source and each terminal state as the sink in a flow network. They consider the reward assigned to each terminal state as an amount of flow and train the policy $P_F$ to distribute the flow from the source to each sink according to the given reward. The flow originating from the source ultimately equals the total flow entering each sink; thus, the total flow $Z$ is defined as follows:

$$Z = \sum_{x \in \chi} R(x). \tag{2}$$

Moreover, with the definition of $Z$ in Equation 2, the formulation of the forward policy, trajectory, and reward can be unified as $R(x) = Z \sum_{\tau=(s_0 \to \cdots \to s_n = x)} P_F(\tau)$. It should be noted that there may be multiple trajectories that lead to a given target object $x$.

The trajectory balance loss (Malkin et al., 2022) was employed to train the model to estimate the flow along a single trajectory $\tau$. The training is guided by the flow consistency constraint, which ensures that the forward flow from the initial state $s_0$ to the terminal state $s_n$ following the forward policy $P_F(\tau)$ and the backward flow from the terminal state $s_n$ to the initial state $s_0$ following the backward policy $P_B(\tau)$ have equal magnitudes. As the forward flow is $ZP_F(\tau)$ and the backward flow is $R(x)P_B(\tau)$, the trajectory balance loss $L_{TB}(\tau)$ that unifies them can be expressed as follows:

$$L_{TB}(\tau) = \left[ \log \frac{Z_\theta \prod_{i=0}^{n-1} P_F(s_{i+1}|s_i;\theta)}{R(x) \prod_{i=0}^{n-1} P_B(s_i|s_{i+1};\theta)} \right]^2. \tag{3}$$

The quantity of flow $Z$ at the initial state $s_0$ is parameterized by a learnable scalar parameter $Z_\theta$, and the forward and backward policies are parameterized as $P_F(-|-;\theta)$ and $P_B(-|-;\theta)$, respectively.

## 3.2 DISCRIMINATOR OF GAN FRAMEWORK

The GAN framework consists of a generator model for capturing the data distribution, and a discriminator model for estimating the probability that a given sample originates from the training data versus from the generator. In the vanilla GAN, the discriminator is trained using Kullback-Leibler divergence (KL divergence) and is designed as a binary classifier to determine whether a given data point is real or synthetic. The generator $G$ and discriminator $D$ are trained using the minimax framework based on the following value function $V(G, D)$,

$$\min_G \max_D V(D, G) = \mathbb{E}_{x \sim p_{data}(x)}[\log D(x)] + \mathbb{E}_{z \sim p_z(z)}[\log(1 - D(G(z)))], \tag{4}$$

where $p_{data}$ and $p_z$ represent the distribution of real data, and latent variable $z$, respectively. The generator $G$ learns the data distribution indirectly through the discriminator $D$ by adversarial training.

## 4 TABGFN

The process of constructing objects in GFlowNets is analogous to the process of generating tabular data instances by determining each feature sequentially. GFlowNets estimate the possible actions to be taken from the current state as a conditional distribution and proceed to the next state through stochastic sampling. Similarly, TabGFN estimates the conditional distribution over the current feature and decides the next feature. The generation process of TabGFN represents the joint distribution of features as a chain of conditional distributions, as expressed in Equation 1.

### 4.1 STATE REPRESENTATION

The state of TabGFN represents not only the determination status of the features but also the categories of the determined features. In TabGFN, each categorical feature is represented as a one-hot

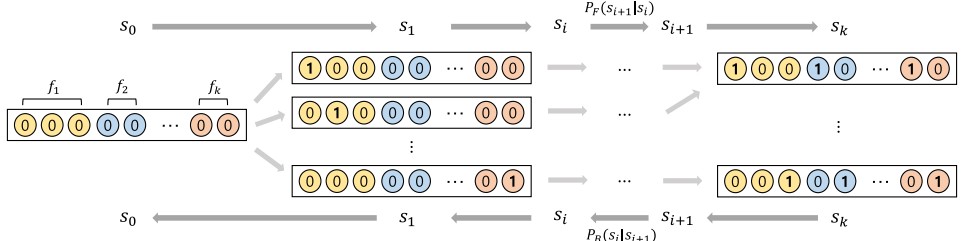

Figure 2: Trajectory sampling process of TabGFN. Forward trajectory sampling starts at initial state $s_0$ and proceeds according to forward policy $P_F$. In each state transition, a feature $f$ and the corresponding category are sampled. The generation process ends once it reaches the terminal state $s_k$. Backward trajectory sampling starts from the given data and proceeds over backward policy $P_B$ until it reaches the initial state $s_0$.

vector, and undetermined features are represented as zero-filled vectors. A feature $f$ with cardinality $C$, and category $c$ is represented as follows:

$$f = [\delta_1, \cdots, \delta_C], \text{ where } \delta_i = \begin{cases} 1, & \text{if } i = c \\ 0, & \text{if } i \neq c \text{ or } c \text{ is undefined} \end{cases}. \tag{5}$$

Therefore, a state consisting of $k$ features is represented as follows: $s = [f_1, \cdots, f_k]$. It should be noted that the initial state $s_0$ is a distinct state in which all features are undetermined, as illustrated in Figure 2. GFlowNets handle a discrete action space; thus, TabGFN requires the discretization of continuous features.

## 4.2 GENERATION PROCESS

The generation process of TabGFN starts from the initial state $s_0$ and involves a series of action sampling procedures to select features and their corresponding categories as illustrated in Figure 2. Importantly, each step (state transition) explicitly occurs according to the conditional distribution estimated through the forward policy $P_F$ as expressed in Equation 1. Given the determined features, the probabilities of the subsequent features and categories are explicitly provided. This sequential feature determination process is a key feature of TabGFN, providing traceability in its generation process. This traceability is consistently available even when using the backward policy to sample trajectories from the given data.

## 4.3 REWARDS

In GFlowNets, the reward represents the unnormalized probability associated with object generation. Therefore, the reward function of TabGFN induces the policy to learn the distribution of the real dataset.

WGAN-GP proposes a training method that addresses the training instability of the vanilla GAN by replacing the KL divergence with the Wasserstein distance. The objective function of WGAN-GP is as follows:

$$L_{critic} = \mathbb{E}_{x \sim \mathbb{P}_r}[D_\phi(x)] - \mathbb{E}_{\tilde{x} \sim \mathbb{P}_g}[D_\phi(\tilde{x})] + \lambda \mathbb{E}_{\hat{x} \sim \mathbb{P}_{\hat{x}}}[(||\nabla_{\hat{x}} D_\phi(\hat{x})||_2 - 1)^2], \tag{6}$$

where $\mathbb{P}_r$ represents the distribution of real data, and $\mathbb{P}_g$ denotes those of generated data. WGAN-GP introduces a gradient penalty, enforcing the Lipschitz constraint. Here, $\mathbb{P}_{\hat{x}}$ in the gradient penalty term refers to sampling along straight lines between samples from the data distributions $\mathbb{P}_r$ and $\mathbb{P}_g$.

In this context, the critic network $D_\phi$ is trained by maximizing $L_{critic}$ for assessing the quality of a given sample. Specifically, the output of the critic network $D_\phi$ is a numerical score correlated with the quality of the given sample (Gulrajani et al., 2017). TabGFN uses the critic network $D_\phi$ as a reward function $R(x; \phi)$ during training; thus, the policy of TabGFN tends to generate high-quality samples.

---

**Algorithm 1** Critic-GFN joint training of TabGFN

---

**Require:** Training dataset $x_t$
1: **repeat**
2:      $c \sim Bernoulli(\alpha)$
3:      **if** $c = 1$ **then**
4:          Sample forward trajectory, $\tau \sim P_F(s_0)$                   ▷ train from real data
5:      **else**
6:          Sample backward trajectory from data, $\tau \sim P_B(x_t)$       ▷ train from generated data
7:      **end if**
8:      Update $\theta$ with $L_{subTB}(\tau)$                              ▷ train the policy network
9:      Sample forward trajectory $\tau \sim P_F(s_0)$
10:     Update $\phi$ with $L_{critic}(x_t, x(\tau))$, where $x(\tau)$ denotes the terminal state of $\tau$    ▷ train the reward(critic) network
11: **until** some convergence conditions

---

## 4.4 TRAINING

Madan et al. (2023) generalized the trajectory balance loss (Equation 3) to facilitate training with partial trajectories. Considering a complete trajectory $\tau = (s_0 \rightarrow \cdots \rightarrow s_n)$, a partial trajectory denoted by $\tau_{i:j} = (s_i \rightarrow \cdots \rightarrow s_j)$, where $0 \leq i < j \leq n$. The intermediate states $s_i$ and $s_j$ act as the source and sink, respectively, of the partial trajectory. To employ the trajectory balance constraints, the flow at the intermediate states is estimated with $F(-;\theta)$. Thus, the trajectory balance loss is defined as follows:

$$L_{subTB}(\tau_{i:j}) = \left[ \log \frac{F(s_i;\theta) \prod_{t=i}^{j-1} P_F(s_{t+1}|s_t;\theta)}{F(s_j;\theta) \prod_{t=i}^{j-1} P_B(s_t|s_{t+1};\theta)} \right]^2 . \tag{7}$$

The flow quantities at source $Z_\theta$ and sink $R(x)$, in Equation 3, are replaced by $F(s_i)$ and $F(s_j)$, respectively. During training, if a complete trajectory is available, then $\binom{n+1}{2}$ sub-trajectories can be obtained. To utilize sub-trajectories of varying lengths for training, weights are assigned to each sub-trajectory. The objective function is defined as follows:

$$L = \frac{\sum_{0 \leq i < j \leq n} \lambda^{j-i} L_{subTB}(\tau_{i:j})}{\sum_{0 \leq i < j \leq n} \lambda^{j-i}}, \tag{8}$$

where $\lambda$ is a hyperparameter that assigns weights to partial trajectories. Training from partial trajectories enables fast convergence.

The training of TabGFN follows a joint training framework that alternates between learning the policy network and critic network, as described in Algorithm 1. The policy network is updated with Equations 7, and 8 using the sub-trajectory balance loss. For effective learning of a state space, TabGFN employs two strategies for trajectory sampling: backward and forward trajectory sampling. Backward trajectory sampling obtains the trajectory using the backward policy of TabGFN. A complete trajectory is obtained starting from terminal state $x$, a given data point, and ends with the initial state $s_0$. Forward trajectory sampling uses the forward policy and starts with the initial state $s_0$ in an active learning manner. Learning from these complementary trajectories leads to an effective approximation of $P_F(\tau;\theta)$ of $x$ and its surroundings in the state space. The critic network is updated with Equation 6 of WGAN-GP which learns from both real and synthetic data. Figure 3 provides an overview of the training scheme, which is motivated by research by Zhang et al. (2022).

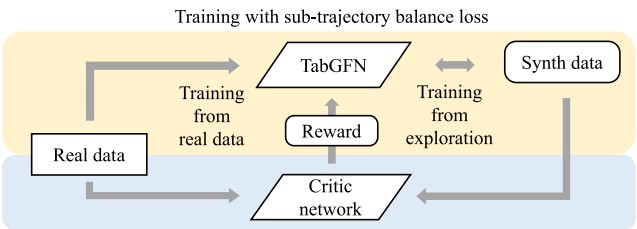

Figure 3: Overview of the training scheme of TabGFN. The yellow shaded area depicts the training process for TabGFN, while the blue shaded area depicts the training process for the critic network. TabGFN is trained using both real and synthetic data generated by the policy.

## 5 EXPERIMENTS

### 5.1 SYNTHESIS TASK

#### 5.1.1 DATASETS

We conducted benchmark tests using 26 datasets, including 10 simulated and 16 real-world datasets. The simulated datasets consisted of four rule-generated datasets and six Bayesian-network-generated datasets. Bayesian-network-generated datasets are obtained by the structures from "bnlearn" (https://www.bnlearn.com/bnrepository/). The real-world datasets were obtained from the UCI repository (https://archive.ics.uci.edu/) and Kaggle (https://www.kaggle.com/). Among them, eight datasets consisted solely of categorical features, while eight consisted of both continuous and categorical features. The details of the datasets are provided in the appendix.

#### 5.1.2 BASELINES

As baselines, we selected two GAN-based approaches, **CTGAN** and **GANBLR**, and a diffusion-based approach, **TabDDPM**. GANBLR uses a Bayesian network as a generator. TabDDPM has achieved SOTA performance.

#### 5.1.3 EVALUATION

The **Machine learning efficacy test**, proposed by Xu et al. (2019), is a crucial evaluation metric for assessing the quality of generated datasets and is widely used to assess tabular data generation. The evaluation methodology is as follows: 1) divide the real dataset into training and test sets; 2) train the generative model on the training set and generate a synthetic dataset of the same size as the training set; 3) train a predictive model, such as XGBoost, on the synthetic data; and 4) evaluate the predictive performance using the test set. If the generative model successfully captures the data distribution, the performance of the predictive model will be comparable to or even surpass that of a model trained on the training set. We generated five sets of synthetic data from each generative model and averaged the machine learning efficacy scores. The macro-averaged F1-score was used as the evaluation metric, and the results are presented in Table 1.

In addition, we proposed a **sample diversity test**, to assess the diversity of the generated samples. High-quality synthetic data should include a wide variety of data samples while avoiding duplicate instances. The sample diversity test calculates the ratio of unique samples in the synthetic data and assigns weights based on the quality of the data. These weights are designed to discern diversity from randomness found in untrained generative models, by calculating the ratio of the machine learning efficacy scores between the synthetic and real data. The sample diversity scores are provided in parentheses in Table 1.

#### 5.1.4 RESULTS ANALYSIS

We evaluated TabGFN, TabDDPM, GANBLR, and CTGAN on various simulated and real-world datasets. On rule-generated datasets, such as balance_scale, car, and monk, TabGFN achieves high performance, while the other methods achieved subpar performance. The rule-generated data have clear causal relationships in their generation, which TabGFN effectively learned. In contrast, the performance of TabGFN was inferior on large-scale Bayesian-network-generated datasets, such as insurance and alarm, possibly due to the critic network. CTGAN, which uses a critic network similar to that of TabGFN, also achieved unsatisfactory results, suggesting that the critic network may struggle to learn certain data distributions. Another possibility lies in the predictive performance of the model. The subpar performance observed even when trained on real data indicates that the predictive model may not have sufficiently captured correlations between features.

On real-world datasets, TabGFN achieved comparable performance to that of TabDDPM, even on datasets including continuous features. While TabDDPM achieved the highest performance by precisely modeling the joint distribution using DDPM, TabGFN learned the conditional distribution and the proper dependence order between features in the form of a Markov chain through backward and forward state exploration.

Table 1: Machine learning efficacy and diversity test results. REAL denotes results obtained by training on the training set and evaluating on the test set. The best score is highlighted in bold, while the diversity scores are in parentheses.

| Data | REAL | TabGFN | TabDDPM | GANBLR | CTGAN |
|------|------|--------|---------|--------|-------|
| balance_scale | 0.636 (1.000) | **0.838** (0.918) | 0.686 (0.426) | 0.581 (0.450) | 0.585 (0.487) |
| car | 0.935 (1.000) | **0.923** (0.649) | 0.271 (0.247) | 0.780 (0.483) | 0.335 (0.280) |
| monk1 | 0.972 (1.000) | **0.980** (0.590) | 0.858 (0.405) | 0.811 (0.603) | 0.702 (0.575) |
| monk2 | 0.819 (1.000) | **0.883** (0.656) | 0.521 (0.511) | 0.628 (0.519) | 0.532 (0.511) |
| monk3 | 0.958 (1.000) | **0.990** (0.599) | 0.957 (0.660) | 0.967 (0.794) | 0.935 (0.814) |
| sachs | 0.737 (0.450) | 0.757 (0.567) | **0.766** (0.389) | 0.741 (0.980) | 0.706 (0.517) |
| asia | 0.916 (0.202) | 0.916 (0.314) | **0.917** (0.162) | **0.917** (0.150) | 0.916 (0.233) |
| child | 0.815 (0.737) | **0.824** (0.909) | 0.817 (0.677) | 0.791 (0.970) | 0.677 (0.584) |
| insurance | 0.555 (0.911) | 0.450 (0.810) | **0.557** (0.899) | 0.515 (0.928) | 0.483 (0.838) |
| alarm | 0.741 (0.636) | 0.672 (0.886) | **0.736** (0.540) | 0.646 (0.871) | 0.585 (0.530) |
| mushroom | 1.000 (1.000) | 0.984 (0.979) | **1.000** (0.713) | 0.979 (0.979) | 0.989 (0.698) |
| nursery | 1.000 (1.000) | **0.958** (0.629) | 0.922 (0.187) | 0.915 (0.522) | 0.675 (0.407) |
| primary_tumor | 0.442 (0.919) | **0.366** (0.824) | 0.204 (0.447) | 0.303 (0.485) | 0.351 (0.715) |
| soybean | 0.931 (0.989) | 0.898 (0.948) | **0.921** (0.614) | 0.828 (0.888) | 0.886 (0.895) |
| spect_heart | 0.567 (0.813) | **0.674** (1.178) | 0.632 (0.607) | 0.567 (0.965) | 0.614 (0.853) |
| breast_cancer | 0.578 (0.959) | **0.702** (1.192) | 0.620 (0.634) | 0.591 (1.023) | 0.627 (0.926) |
| house_vote | 0.942 (0.735) | 0.971 (0.839) | 0.968 (0.544) | **0.974** (1.027) | 0.957 (0.748) |
| lymphography | 0.420 (1.000) | **0.545** (1.270) | 0.415 (0.697) | 0.400 (0.953) | 0.440 (0.992) |
| glioma | 0.809 (0.998) | **0.842** (0.972) | **0.842** (1.044) | 0.816 (1.008) | 0.809 (0.999) |
| iris | 0.956 (0.981) | **0.955** (0.888) | 0.947 (0.991) | 0.700 (0.583) | 0.831 (0.870) |
| mine | 0.469 (1.000) | **0.571** (0.964) | 0.554 (1.108) | 0.161 (0.325) | 0.321 (0.684) |
| adult | 0.814 (0.999) | 0.774 (0.935) | **0.788** (0.938) | 0.709 (0.870) | 0.776 (0.953) |
| cardio | 0.728 (1.000) | 0.639 (0.141) | **0.729** (1.001) | 0.482 (0.660) | 0.707 (0.971) |
| churn | 0.751 (1.000) | **0.735** (0.973) | 0.731 (0.967) | 0.539 (0.718) | 0.689 (0.918) |
| diabetes | 0.682 (1.000) | 0.701 (1.028) | 0.707 (1.064) | 0.523 (0.767) | **0.719** (1.054) |
| loan | 0.988 (1.000) | **0.880** (0.891) | 0.818 (0.499) | 0.477 (0.483) | 0.872 (0.883) |
| overall mean | 0.775 (0.897) | **0.786** (0.829) | 0.726 (0.653) | 0.667 (0.731) | 0.682 (0.728) |

In the sample diversity test, TabDDPM achieved the lowest average performance, while TabGFN achieved the highest performance. Considering that some of the data included continuous features and TabGFN can only generate discretized value, this result is particularly noteworthy. These results demonstrate that the diversity of GFlowNets is well represented in TabGFN.

## 5.2 TRACEABILITY

The generation process of TabGFN provides traceability. At each step, the forward policy $P_F$ of TabGFN estimates the sampling probability of undetermined features and their respective categories. Figure 4 displays the forward distributions estimated in the generation process by TabGFN trained on the balance_scale dataset. The balance_scale dataset consisted of five features. $C$ denotes the balance of the scale (L: left, B: balance, R: right), while $D_L$, $D_R$, $W_L$, and $W_R$ denote the state of each arm. $D$ and $W$ represent the distance and weight respectively, with subscript $L$ and $R$ denoting the left and right arm, respectively. The data generated by TabGFN resulted in $C = B$, $W_L = 1$, $D_L$=4, $W_R = 2$, and $W_D = 2$. The weights and distances on both sides of the scale are independent, but the balance of the scale is dependent on the state of both arms. This dependence was represented in the generation process. The probability of generating $C$ increased towards the end of the generation process, as displayed in Figure 4. State $s_3$ had determined features $W_L = 1$, $D_L = 4$, and $W_R = 2$. The undetermined feature $D_R$ is independent of the other variables; thus, the probability of its five categories appears quite uniform (see row $P_F(s_4|s_3)$). However, the categorical distribution for $C$ in the same row did not follow this pattern. The case of $C = L$ occurred only when $D_R = 1$, while the opposite was possible when $D_R$ was 3, 4, and 5. The probability of $C = R$ was approximately three times that of $C = L$. The feature sampler determined $D_R$ as 2; thus, the probability of $C = B$ estimated in $P_F(s_5|s_4)$ significantly increased.

Additionally, global feature dependencies can be inferred by analyzing the order of feature generation. For instance, upon generating a sufficient number of balance scale data instances and averaging the feature creation sequences, the following order emerged: $W_L = 1.61$, $D_L = 1.66$, $W_R = 1.68$, $D_R = 1.58$, and $C = 3.48$. The features related to the state of each arm of the scale exhibited almost identical generation orders, while $C$ representing the state of the scale was determined last. This result is consistent with the intuitive understanding that the weight and distance of each scale arm are independent, whereas the state of the scale is dependent on both the weight and distance of

| | C | | | $W_I = 1$ | | | | | $D_I = 4$ | | | | | $W_R = 2$ | | | | | $D_R = 2$ | | | | |
| | L | B | R | 1 | 2 | 3 | 4 | 5 | 1 | 2 | 3 | 4 | 5 | 1 | 2 | 3 | 4 | 5 | 1 | 2 | 3 | 4 | 5 |
|---|---|---|---|---|---|---|---|---|---|---|---|---|---|---|---|---|---|---|---|---|---|---|---|
| $P_F(s_1\|s_0)$ | 0.008 | 0.006 | 0.009 | 0.036 | 0.054 | 0.053 | 0.052 | 0.051 | 0.037 | 0.052 | 0.054 | 0.051 | 0.047 | 0.040 | **0.057** | 0.054 | 0.051 | 0.044 | 0.040 | 0.051 | 0.056 | 0.051 | 0.046 |
| $P_F(s_2\|s_1)$ | 0.020 | 0.006 | 0.016 | 0.057 | 0.068 | 0.065 | 0.065 | 0.064 | 0.059 | 0.067 | 0.067 | **0.064** | 0.062 | 0.000 | 0.000 | 0.000 | 0.000 | 0.000 | 0.059 | 0.064 | 0.067 | 0.066 | 0.064 |
| $P_F(s_3\|s_2)$ | 0.065 | 0.005 | 0.024 | **0.103** | 0.095 | 0.085 | 0.083 | 0.084 | 0.000 | 0.000 | 0.000 | 0.000 | 0.000 | 0.000 | 0.000 | 0.000 | 0.000 | 0.000 | 0.085 | 0.083 | 0.091 | 0.096 | 0.100 |
| $P_F(s_4\|s_3)$ | 0.047 | 0.007 | 0.146 | 0.000 | 0.000 | 0.000 | 0.000 | 0.000 | 0.000 | 0.000 | 0.000 | 0.000 | 0.000 | 0.000 | 0.000 | 0.000 | 0.000 | 0.000 | 0.178 | **0.186** | 0.154 | 0.144 | 0.138 |
| $P_F(s_5\|s_4)$ | 0.050 | **0.873** | 0.077 | 0.000 | 0.000 | 0.000 | 0.000 | 0.000 | 0.000 | 0.000 | 0.000 | 0.000 | 0.000 | 0.000 | 0.000 | 0.000 | 0.000 | 0.000 | 0.000 | 0.000 | 0.000 | 0.000 | 0.000 |

Figure 4: Example of the generation process, where TabGFN has trained to generate balance scale data. Each column represents a feature and its corresponding categories. Red represents high probability, while green indicates low probability. Each row represents a feature sampling step.

the arms. Based on these examples, we discovered that the traceable process of determining feature values in TabGFN effectively reflects the dependencies among features. This traceability implies TabGFN's capability to serve as an interpretable model.

# 6 DISCUSSION

We attribute the promising results of TabGFN to its ability to capture complex feature dependencies. In comparison, CTGAN relies on only a single depth of feature dependencies when building the conditional generator, while GANBLR's KDB typically uses one or two. These limited configurations may not sufficiently capture the intricate relationships between features. In contrast, TabGFN can learn conditional distributions without depth constraints, providing a more sophisticated representation. TabDDPM also achieves excellent performance due to its deep representation of joint probability distributions; however, it struggles to extract explicit relations due to its implicit nature. In contrast, TabGFN facilitates explicit feature determination and provides traceability, which is crucial for high-stakes inferences. This traceability arises from the clear representation of feature determination probabilities, providing insights into the generation process and highlighting the feature dependencies. This traceability may be a foundational step toward creating an interpretable model.

Nevertheless, TabGFN has limitations. GFlowNets are limited to discrete action space; consequently, TabGFN is only capable of handling tabular data composed of discretized features. However, research on GFlowNets for continuous action spaces, as reported by Lahlou et al. (2023), may lead to extensions to heterogeneous data types. Furthermore, Zaidi et al. (2020) reported that discretization of continuous features can actually enhance performance in crucial tasks involving tabular data, such as classification.

More importantly, the policy does not converge to the real data distribution because the critic network does not provide an exact likelihood of the real data distribution. In addition, GANs cannot guarantee to approximate the data distribution due to the mode collapse (Arora et al., 2018). However, in practice, the policy may asymptotically approximate the data distribution by learning to generate samples that obtain high scores from the critic network.

# 7 CONCLUSION

In this paper, we propose an effective technique for generating tabular data using GFlowNets. Specifically, we treat the generation process of GFlowNets as a sequential feature determination process for tabular data instances. The process ensures traceability while generating high-quality data by learning the dependencies and order of features. TabGFN achieves performance comparable to that of SOTA techniques such as TabDDPM on various benchmarks. These results demonstrate the capability of TabGFN for data generation or augment. In future work, we plan to conduct a theoretical analysis of TabGFN with the critic network. We also intend to extend TabGFN to handle both continuous and categorical features simultaneously. Additionally, we would develop an interpretation method for TabGFN that exploits its traceability.

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

# A DATASETS

Table 2 contains detailed information on the 26 benchmark datasets consisting of 10 simulated and 16 real-world datasets. The balance_scale, car, and monk datasets are generated according to specific rules, while the sachs, asia, child, insurance, and alarm datasets are generated through Bayesian networks obtained from the "bnlearn" repository (`https://www.bnlearn.com/bnrepository/`). The 16 real-world datasets are obtained from the UCI repository (`https://archive.ics.uci.edu/`) and Kaggle (`https://www.kaggle.com/`), and they are in various compositions: some datasets consist solely of categorical features, while others include a mix of both continuous and categorical features. In cases where the original dataset did not provide a train-test split, we divided the data into training and testing sets using a 7:3 ratio, stratified by the target feature.

Table 2: Summary of benchmark datasets. # Con and # Cat refer to the number of continuous and categorical features respectively. # Cls denotes the number of target classes. # Train and # Test indicate the sizes of training and test datasets. State dims represents the dimension of the state in TabGFN.

| Data | # Con | # Cat | # Cls | # Train | # Test | State dims |
|---|---|---|---|---|---|---|
| balance_scale | 0 | 5 | 3 | 437 | 188 | 23 |
| car | 0 | 7 | 4 | 1209 | 519 | 25 |
| monk1 | 0 | 7 | 2 | 124 | 432 | 19 |
| monk2 | 0 | 7 | 2 | 169 | 432 | 19 |
| monk3 | 0 | 7 | 2 | 122 | 432 | 19 |
| sachs | 0 | 11 | 3 | 2100 | 900 | 33 |
| asia | 0 | 8 | 2 | 84 | 36 | 16 |
| child | 0 | 20 | 6 | 21000 | 9000 | 60 |
| insurance | 0 | 27 | 4 | 7000 | 3000 | 88 |
| alarm | 0 | 37 | 3 | 7000 | 3000 | 105 |
| mushroom | 0 | 22 | 2 | 3950 | 1694 | 99 |
| nursery | 0 | 9 | 4 | 9070 | 3888 | 31 |
| primary_tumor | 0 | 18 | 12 | 86 | 37 | 49 |
| soybean | 0 | 36 | 15 | 266 | 296 | 112 |
| spect_heart | 0 | 23 | 2 | 80 | 187 | 46 |
| breast_cancer | 0 | 10 | 2 | 193 | 84 | 43 |
| house_vote | 0 | 17 | 2 | 162 | 70 | 34 |
| lymphography | 0 | 18 | 3 | 102 | 44 | 58 |
| glioma | 1 | 23 | 2 | 587 | 252 | 58 |
| iris | 4 | 1 | 3 | 105 | 45 | 43 |
| mine | 3 | 1 | 5 | 236 | 102 | 35 |
| adult | 6 | 9 | 2 | 21113 | 9049 | 160 |
| cardio | 5 | 7 | 2 | 49000 | 21000 | 66 |
| churn | 6 | 5 | 2 | 7000 | 3000 | 71 |
| diabetes | 8 | 1 | 2 | 537 | 231 | 82 |
| loan | 12 | 2 | 2 | 6704 | 2874 | 129 |

# B EXPERIMENTAL DETAILS

To ensure the reproducibility of the experimental results, we fixed the random seed for Python packages including Python random, PyTorch, TensorFlow, Numpy, and XGBoost at 7. For the comparison algorithms, we utilized the code for TabDDPM from `https://github.com/yandex-research/tab-ddpm`, for GANBLR from `https://github.com/tulip-lab/open-code/tree/develop/GANBLR`, and for CTGAN from `https://github.com/sdv-dev/CTGAN`. The hyperparameters applied in the experiments are described in Table 3. The term $n$-bins refers to the number of bins applied when discretizing continuous variables and is relevant for TabGFN and GANBLR, which can only handle discretized features. The 'Network' section specifies the dimensions (dims) and depth of the MLP layers, and these parameters were consistently applied to the architecture of all neural networks used in the experiments. The $\lambda$ in "$L_{critic}$" corresponds to the gradient penalty term in Equation 6. In the section "$L_{subTB}$", the $j - i$ indicates the maximum length of the partial trajectory; where $\lambda$ represents the weight for each partial trajectory (refers to Eqation 8). The "Train" section describes the learning rate (LR), mini-batch size (batch), and number of epochs (epoch).

Table 3: Summary of the hyperparameter configurations during experiments.

| Data | Preprocess $n$-bins | Network dims | depth | $L_{critic}$ $\lambda$ | $L_{subTB}$ $j-i$ | $L_{subTB}$ $\lambda$ | LR | Train batch | epoch |
|---|---|---|---|---|---|---|---|---|---|
| balance_scale | - | 100 | 5 | 1 | 5 | 0.8 | 0.001 | 10k | 5k |
| car | - | 100 | 3 | 1 | 7 | 0.8 | 0.005 | 10k | 5k |
| monk1 | - | 100 | 3 | 1 | 7 | 0.8 | 0.001 | 10k | 5k |
| monk2 | - | 100 | 5 | 1 | 7 | 0.8 | 0.001 | 10k | 5k |
| monk3 | - | 100 | 5 | 1 | 7 | 0.8 | 0.001 | 10k | 10k |
| sachs | - | 100 | 3 | 1 | 11 | 0.8 | 0.005 | 10k | 5k |
| asia | - | 100 | 3 | 1 | 8 | 0.8 | 0.005 | 10k | 5k |
| child | - | 200 | 7 | 1 | 20 | 0.8 | 0.001 | 10k | 15k |
| insurance | - | 100 | 3 | 1 | 27 | 0.8 | 0.001 | 10k | 10k |
| alarm | - | 200 | 7 | 1 | 20 | 0.8 | 0.001 | 10k | 10k |
| mushroom | - | 100 | 5 | 1 | 23 | 0.8 | 0.001 | 10k | 10k |
| nursery | - | 100 | 7 | 1 | 9 | 0.8 | 0.001 | 10k | 5k |
| primary_tumor | - | 100 | 5 | 1 | 27 | 0.8 | 0.001 | 10k | 10k |
| soybean | - | 200 | 7 | 1 | 27 | 0.8 | 0.0005 | 10k | 35k |
| spect_heart | - | 100 | 5 | 1 | 23 | 0.8 | 0.001 | 10k | 15k |
| breast_cancer | - | 100 | 5 | 1 | 10 | 0.8 | 0.001 | 10k | 15k |
| house_vote | - | 200 | 7 | 1 | 17 | 0.8 | 0.001 | 10k | 15k |
| lymphography | - | 300 | 5 | 1 | 18 | 0.8 | 0.0005 | 10k | 15k |
| glioma | 10 | 100 | 3 | 1 | 24 | 0.8 | 0.005 | 10k | 5k |
| iris | 10 | 100 | 5 | 1 | 5 | 0.8 | 0.001 | 10k | 15k |
| mine | 10 | 100 | 3 | 1 | 4 | 0.8 | 0.001 | 10k | 15k |
| adult | 10 | 200 | 5 | 1 | 15 | 0.8 | 0.0005 | 10k | 10k |
| cardio | 10 | 100 | 5 | 1 | 12 | 0.8 | 0.001 | 10k | 10k |
| churn | 10 | 100 | 5 | 1 | 11 | 0.8 | 0.001 | 10k | 15k |
| diabetes | 10 | 100 | 5 | 1 | 9 | 0.8 | 0.001 | 10k | 15k |
| loan | 10 | 100 | 5 | 1 | 14 | 0.8 | 0.001 | 10k | 15k |

## B.1 MACHINE LEARNING EFFICACY TEST

Table 4 presents the results from the machine learning efficacy test, including the mean and standard deviation across five repeated experiments.

Table 4: Machine learning efficacy test results including standard deviations.

| Data | TabGFN | TabDDPM | GANBLR | CTGAN |
|---|---|---|---|---|
| balance_scale | $0.838 \pm 0.025$ | $0.686 \pm 0.033$ | $0.581 \pm 0.019$ | $0.585 \pm 0.033$ |
| car | $0.923 \pm 0.022$ | $0.271 \pm 0.023$ | $0.780 \pm 0.016$ | $0.335 \pm 0.033$ |
| monk1 | $0.980 \pm 0.008$ | $0.858 \pm 0.035$ | $0.811 \pm 0.071$ | $0.702 \pm 0.029$ |
| monk2 | $0.883 \pm 0.008$ | $0.521 \pm 0.030$ | $0.628 \pm 0.015$ | $0.532 \pm 0.030$ |
| monk3 | $0.990 \pm 0.004$ | $0.957 \pm 0.027$ | $0.967 \pm 0.011$ | $0.935 \pm 0.019$ |
| sachs | $0.757 \pm 0.013$ | $0.766 \pm 0.006$ | $0.741 \pm 0.006$ | $0.706 \pm 0.025$ |
| asia | $0.916 \pm 0.000$ | $0.917 \pm 0.000$ | $0.917 \pm 0.000$ | $0.916 \pm 0.000$ |
| child | $0.824 \pm 0.002$ | $0.817 \pm 0.003$ | $0.791 \pm 0.004$ | $0.677 \pm 0.005$ |
| insurance | $0.450 \pm 0.017$ | $0.557 \pm 0.015$ | $0.515 \pm 0.021$ | $0.483 \pm 0.015$ |
| alarm | $0.672 \pm 0.004$ | $0.736 \pm 0.005$ | $0.646 \pm 0.045$ | $0.585 \pm 0.009$ |
| mushroom | $0.984 \pm 0.008$ | $1.000 \pm 0.000$ | $0.979 \pm 0.023$ | $0.989 \pm 0.004$ |
| nursery | $0.958 \pm 0.006$ | $0.922 \pm 0.005$ | $0.915 \pm 0.004$ | $0.675 \pm 0.013$ |
| primary_tumor | $0.366 \pm 0.040$ | $0.204 \pm 0.077$ | $0.303 \pm 0.082$ | $0.351 \pm 0.046$ |
| soybean | $0.898 \pm 0.019$ | $0.921 \pm 0.013$ | $0.828 \pm 0.026$ | $0.886 \pm 0.023$ |
| spect_heart | $0.674 \pm 0.045$ | $0.632 \pm 0.032$ | $0.567 \pm 0.044$ | $0.614 \pm 0.041$ |
| breast_cancer | $0.702 \pm 0.044$ | $0.620 \pm 0.038$ | $0.591 \pm 0.033$ | $0.627 \pm 0.023$ |
| house_vote | $0.971 \pm 0.010$ | $0.968 \pm 0.006$ | $0.974 \pm 0.006$ | $0.957 \pm 0.015$ |
| lymphography | $0.545 \pm 0.084$ | $0.415 \pm 0.061$ | $0.400 \pm 0.055$ | $0.440 \pm 0.042$ |
| glioma | $0.842 \pm 0.017$ | $0.842 \pm 0.017$ | $0.816 \pm 0.048$ | $0.809 \pm 0.030$ |
| iris | $0.955 \pm 0.023$ | $0.947 \pm 0.020$ | $0.700 \pm 0.091$ | $0.831 \pm 0.077$ |
| mine | $0.571 \pm 0.014$ | $0.554 \pm 0.023$ | $0.161 \pm 0.033$ | $0.321 \pm 0.046$ |
| adult | $0.774 \pm 0.004$ | $0.788 \pm 0.004$ | $0.709 \pm 0.027$ | $0.776 \pm 0.003$ |
| cardio | $0.639 \pm 0.001$ | $0.729 \pm 0.002$ | $0.482 \pm 0.049$ | $0.707 \pm 0.003$ |
| churn | $0.735 \pm 0.007$ | $0.731 \pm 0.005$ | $0.539 \pm 0.034$ | $0.689 \pm 0.007$ |
| diabetes | $0.701 \pm 0.031$ | $0.707 \pm 0.023$ | $0.523 \pm 0.077$ | $0.719 \pm 0.019$ |
| loan | $0.880 \pm 0.005$ | $0.818 \pm 0.005$ | $0.477 \pm 0.120$ | $0.872 \pm 0.004$ |

Figure 5 presents the results of the machine learning efficacy test according to the epoch cycle during training. The y-axis of each plot represents the macro-averaged F1 score, while the x-axis represents the epoch cycle. The graph shows an improving trend in the quality of generation as training progresses. For GANBLR, which uses a Bayesian network, the training time per epoch is significantly longer than other algorithms. However, it requires fewer epochs to converge, allowing

us to set a relatively shorter epoch cycle for experimentation. The results of the Machine Learning Efficacy Test are based on the epoch that demonstrated the best performance in this graph. The results of other experiments such as the sample diversity test are also based on the data generated at that epoch.

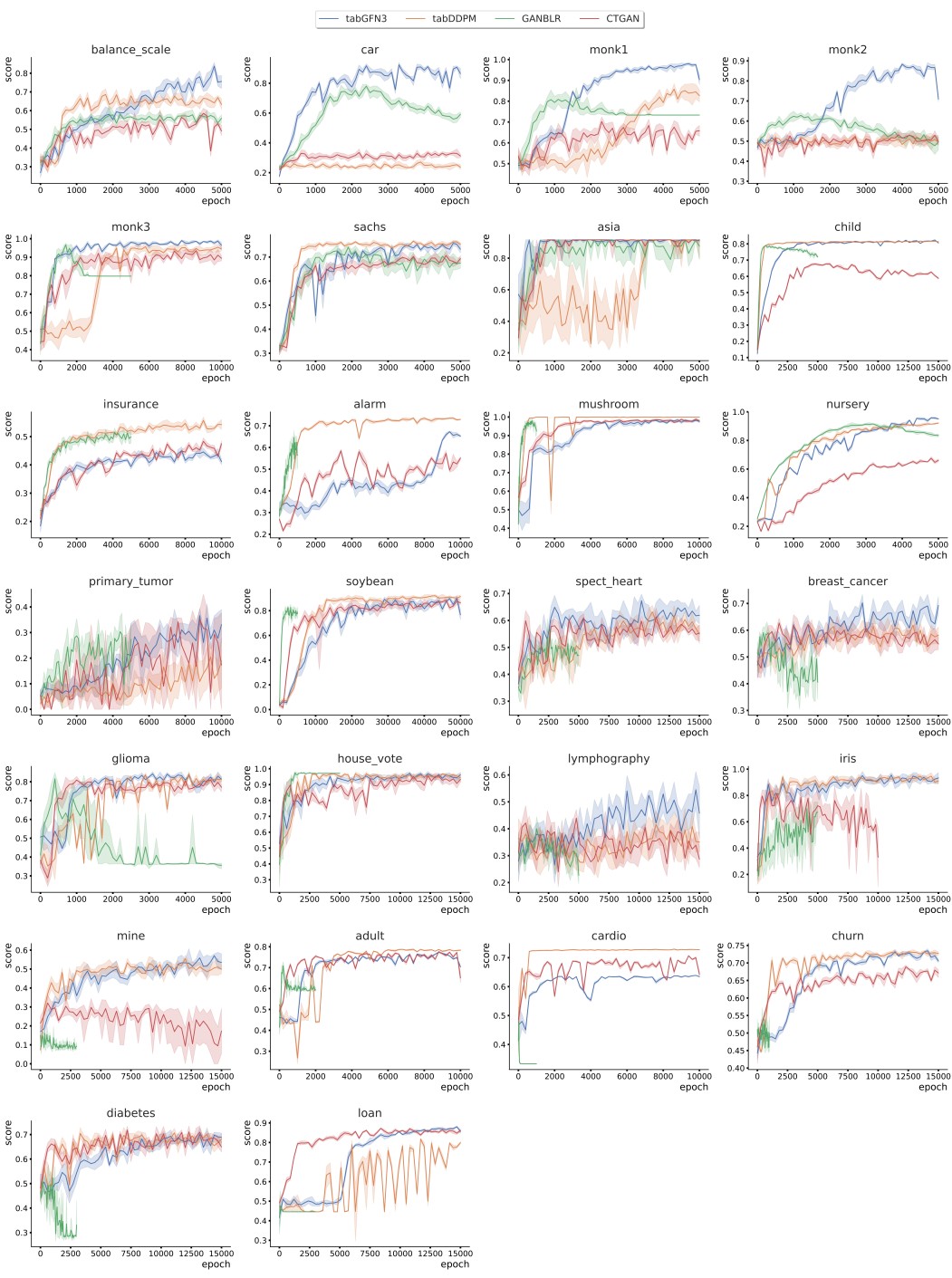

Figure 5: Changes in machine learning efficacy score according to training epochs.

## B.2 SAMPLE DIVERSITY TEST

Table 5 presents the results used in the sample diversity Test, displaying the unweighted ratio of unique samples. The table includes both the average and standard deviation based on five repeated trials.

Table 5: Unweighted sample diversity scores with standard deviations.

| Data | TabGFN | TabDDPM | GANBLR | CTGAN |
|---|---|---|---|---|
| balance_scale | $0.697 \pm 0.012$ | $0.753 \pm 0.008$ | $0.493 \pm 0.006$ | $0.529 \pm 0.009$ |
| car | $0.657 \pm 0.004$ | $0.844 \pm 0.006$ | $0.579 \pm 0.011$ | $0.781 \pm 0.017$ |
| monk1 | $0.586 \pm 0.007$ | $0.674 \pm 0.032$ | $0.723 \pm 0.032$ | $0.797 \pm 0.031$ |
| monk2 | $0.609 \pm 0.012$ | $0.788 \pm 0.022$ | $0.677 \pm 0.021$ | $0.788 \pm 0.035$ |
| monk3 | $0.580 \pm 0.006$ | $0.662 \pm 0.033$ | $0.787 \pm 0.030$ | $0.834 \pm 0.016$ |
| sachs | $0.538 \pm 0.010$ | $0.373 \pm 0.010$ | $0.975 \pm 0.003$ | $0.539 \pm 0.005$ |
| asia | $0.314 \pm 0.021$ | $0.162 \pm 0.031$ | $0.150 \pm 0.021$ | $0.233 \pm 0.012$ |
| child | $0.899 \pm 0.001$ | $0.681 \pm 0.005$ | $1.000 \pm 0.000$ | $0.703 \pm 0.003$ |
| insurance | $1.000 \pm 0.000$ | $0.914 \pm 0.002$ | $1.000 \pm 0.000$ | $0.964 \pm 0.002$ |
| alarm | $0.977 \pm 0.001$ | $0.543 \pm 0.003$ | $1.000 \pm 0.000$ | $0.672 \pm 0.005$ |
| mushroom | $0.995 \pm 0.001$ | $0.713 \pm 0.004$ | $1.000 \pm 0.000$ | $0.706 \pm 0.004$ |
| nursery | $0.657 \pm 0.001$ | $0.716 \pm 0.004$ | $0.570 \pm 0.005$ | $0.603 \pm 0.002$ |
| primary_tumor | $0.995 \pm 0.004$ | $0.844 \pm 0.026$ | $0.707 \pm 0.037$ | $0.900 \pm 0.035$ |
| soybean | $0.983 \pm 0.004$ | $0.637 \pm 0.013$ | $0.998 \pm 0.002$ | $0.941 \pm 0.016$ |
| spect_heart | $0.991 \pm 0.004$ | $0.552 \pm 0.029$ | $0.965 \pm 0.023$ | $0.787 \pm 0.016$ |
| breast_cancer | $0.981 \pm 0.007$ | $0.646 \pm 0.014$ | $1.000 \pm 0.000$ | $0.854 \pm 0.023$ |
| house_vote | $0.814 \pm 0.028$ | $0.526 \pm 0.027$ | $0.994 \pm 0.006$ | $0.737 \pm 0.039$ |
| lymphography | $0.979 \pm 0.009$ | $0.704 \pm 0.029$ | $1.000 \pm 0.000$ | $0.949 \pm 0.033$ |
| glioma | $0.934 \pm 0.006$ | $1.000 \pm 0.001$ | $1.000 \pm 0.000$ | $0.999 \pm 0.001$ |
| iris | $0.888 \pm 0.025$ | $1.000 \pm 0.000$ | $0.796 \pm 0.035$ | $1.000 \pm 0.000$ |
| mine | $0.792 \pm 0.012$ | $0.999 \pm 0.002$ | $0.947 \pm 0.013$ | $1.000 \pm 0.000$ |
| adult | $0.984 \pm 0.001$ | $0.975 \pm 0.001$ | $1.000 \pm 0.000$ | $1.000 \pm 0.000$ |
| cardio | $0.160 \pm 0.001$ | $1.000 \pm 0.000$ | $0.996 \pm 0.000$ | $1.000 \pm 0.000$ |
| churn | $0.994 \pm 0.001$ | $0.996 \pm 0.000$ | $0.999 \pm 0.000$ | $1.000 \pm 0.000$ |
| diabetes | $1.000 \pm 0.001$ | $0.999 \pm 0.001$ | $1.000 \pm 0.000$ | $1.000 \pm 0.000$ |
| loan | $1.000 \pm 0.000$ | $0.707 \pm 0.006$ | $1.000 \pm 0.000$ | $1.000 \pm 0.000$ |

