# OpenReview forum: "TabGFN: Tabular data generation based on GFlowNets"
_ICLR.cc/2024/Conference — ICLR 2024 Conference Withdrawn Submission_

### Official Review · Reviewer_wX9t · 2023-10-26

**Soundness:** 3 good
**Presentation:** 2 fair
**Contribution:** 2 fair
**Rating:** 5
**Confidence:** 3

**Summary:**

They used GFlowNets to simultaneously achieve high-predictive performance and model traceability in synthetic tabular data generation. Additionally, the performance of the model was improved by training a critic network using WGAN-GP. Using these networks, it showed better performance than TabDDPM, a SOTA model.

**Strengths:**

Originality: They are the first to apply GFlowNets to tabular synthesis and demonstrate the traceability of tabular data synthesis, which is a problem with neural network-based models.

Quality: Balance scale dataset is used to provide an appropriate example of traceability.

Clarity: By using appropriate baselines, TabGFN's position in synthetic data generation was well represented. TabDDPM-SOTA model, GANBLR-Interpretable generative model and CT-GAN-Using cretic network.

**Weaknesses:**

1. It is not shown why traceability is important in tabular data generation. In other words, the reasons to use GFlowNets are weak.

2. There is no ablation study. An ablation study is needed in the proposed network.

3. The explanation of the experiment results is insufficient. The meaning of the Diversity scores in Table 1 is unknown.

**Questions:**

1. Why is traceability important in Tabular composition?

2. I am curious about the results of TabGFN without a cretic network.

3. I am curious about what Diversity score=1 means.

4. Also, what does it mean to show better performance than real data in an efficacy test? Shouldn’t it be similar to real data for good performance? For example, if balance_scale is real:0.636, TabGFN:0.838, TabDDPM:0.686, isn't the result of TabDDPM better generated because it is more similar to real?

---

### Official Review · Reviewer_eJ3x · 2023-10-27

**Soundness:** 3 good
**Presentation:** 3 good
**Contribution:** 2 fair
**Rating:** 5
**Confidence:** 2

**Summary:**

This paper proposes a new method for synthetic tabular data generation. Their approach hinges on two existing methods namely the Gflownets and WGAN-DP. Gflownets use the training data to approximate the joint distribution of the features by using a chain of conditional distributions. WGAN-DP is used to assess the quality of each feature generated. The proposed method is then extensively evaluated on many tabular datasets.

**Strengths:**

1. The paper for the most part is well-written and the related work and the gap in the literature are well-explained.
2. The proposed method is evaluated thoroughly.
3. The proposed method performs well relative to the other SOTA and provides insight into the data generation process.

**Weaknesses:**

1. One major weakness of this paper is that the code used for implementing the proposed method and the experiments are not made available through supplementary material.

2. The authors make a few claims about why their proposed methods work better without any justification.
"We attribute the promising results of TabGFN to its ability to capture complex feature dependencies."
"TabGFN can learn conditional distributions without depth constraints, providing a more sophisticated representation"

2. The authors also show that for their diversity metric, their proposed model performs better. However, they provide no justification why this might be the case.

3. In the abstract, you mention that synthetic data generation can be used for privacy preservation of data but no privacy guarantees are shown.

**Questions:**

The questions follow the comments I made on the weaknesses.

1. Can you make the implementation of the proposed method available?
2. Can you justify the claims made about the reasons behind the improved performance by your method?
3. Can you explain why your proposed method performs so much better in terms of the diversity metric?
4. Can you show any privacy guarantees for your proposed method?

---

### Official Review · Reviewer_erTM · 2023-10-31

**Soundness:** 2 fair
**Presentation:** 2 fair
**Contribution:** 2 fair
**Rating:** 5
**Confidence:** 3

**Summary:**

This paper proposes TabGFN which applies GFlowNets, equipped with a WGAN-GP critic as a generative model for tabular data. The method is one of several recent methods leveraging advances in the deep generative models to improve synthetic data generation in tabular settings - which is important in the context the privacy. The authors adopt the GFlowNets framework to model the joint distribution over features. Specifically, the GFlowNet policy generates the data points sequentially, selecting one feature at each step (in any order). The energy function uses to train the GFlowNet is specified the discriminator (critic) of a WGAN-GP, i.e. the GFlowNet is trained to generate candidates that are classified as "real" by the discriminator. In essence the GFlowNet acts as the generator in a standard GAN setup. The authors adopt methodological details from prior work on GFlowNets. The method is evaluated against existing GAN and diffusion approaches in terms of the quality of data generation as well as diversity of generated samples, and analyze the traceability of the policy.

**Strengths:**

* The method successfully leverages the strengths of GFlowNets in a novel, unique application for tabular data generation. The problem and why the framework of GFlowNets is relevant is well motivated.
* The idea of combining the GFlowNet generative policy with a critic from GANs also is quite interesting.
* The empirical results match prior state-of-the-art despite using only discretized categorical features on a variety of benchmarks.

**Weaknesses:**

* A big advantage of the approach that the authors focus on is traceability. I am not sure why this traceability is an advantage at all. The GFlowNet generative policy models a feature selection process but the transition probabilities do not capture any underlying notion of order or importance in the features. The learning procedure simple learns . The interpretation of the transition probabilities as proxies to feature importance seems problematic. Moreover, there is little in terms of motivation for this analysis.
* The diversity metric also seems a bit poorly motivated. It is unclear why diversity matters here at all, since it would be captured implicitly in the ML efficiency results - if the data generated doesn't represent the distribution well (i.e. is diverse) then the accuracy would be poorer.
* Finally, the choice of metrics seems a bit specific. Prior work (eg. TabDDPM) uses other metrics like DCR which seem very important for methods like this.
* Reproducibility: The authors do provide relevant hyperparameters, and details but do not provide the code with the submission (and no mentioned of release).

**Questions:**

* How exactly are the continuous features discretized?
* What is the model selection procedure for the generative models?
* In many applications GFlowNets are trained to model tempered rewards. Is that the case here? If so, what temperature was used? If not, wouldn't it be useful to bias the distribution of candidates from the GFlowNet for faster convergence?

---

### Official Review · Reviewer_a1vn · 2023-10-31

**Soundness:** 2 fair
**Presentation:** 2 fair
**Contribution:** 2 fair
**Rating:** 3
**Confidence:** 5

**Summary:**

The paper is an application of GFlowNet, recently proposed by Bengio et al. in 2021, to generate synthetic tabular data. The authors propose to use Wasserstein GAN with gradient penalty to criticize the quality foundation.

**Strengths:**

- The paper seems to be a good use case for GFlowNets.
- Synthesizing tabular data is important in many applications.

**Weaknesses:**

- Novelty is limited.

- The experimental section is not convincing, and it lacks SOTAs (e.g. GOGGLE, VIME, SubTab, etc.), novel metrics (e.g., Quality, Utility, etc.), and better datasets (such as Adult, etc.).

- The dimensionality of the features is too small, with most cases having fewer than 10 features.

- Using a simulation dataset to generate a simulation doesn't make sense, even for evaluation.

- An ablation study for the stability of training with WGAN is needed.

- The proposed method inherently cannot capture a large and realistic number of features, which is somehow one of the drawbacks of GFlowNets. The authors didn't address this and just use current GFN methods.

- It is not clear to me how independent features will be generated. The proposed method inherently imposes some interactions that might not necessarily hold true for tabular data.

- It is not clear whether ordering the features is something of interest.

- It is not clear what additional benefits GFlowNet can bring compared to RL in this setting.

- Plotting the representation space through both VAE-based methods as well as self-supervised learning, such as VIME, SubTab, SCARF, etc., seems interesting to ensure that the model is not generating trivial samples due to WGAN and GFlowNet.

**Questions:**

- Do the samples need to have the same number of features?

- Is the order of features the same for all samples?

- Additional experiments are required with novel baselines, more realistic datasets, as well as better evaluation metrics. For example, GOGGLE is last year's ICLR paper which was missed.

---

### Author Response · Authors · 2023-11-22
**Withdrawal**

We would like to express our gratitude to the reviewers for dedicating their valuable time and providing helpful review comments. However, we have decided to withdraw our paper submission and focus on improving the paper.